# Lymphatic Invasion Might Be Considered as an Upstaging Factor in N0 and N1 Gastric Cancer **[note 1]**

**DOI:** 10.3390/jcm9051275

**Published:** 2020-04-28

**Authors:** Won Hyuk Choi, Min Jeong Kim, Jun Ho Park, Jin Gu Kang, Seung In Seo, Hak Yang Kim, Woon Geon Shin

**Affiliations:** 1Department of Surgery, Kangdong Sacred Heart Hospital, Hallym University College of Medicine, Seoul 05355, Korea; 2Department of Internal Medicine, Kangdong Sacred Heart Hospital, Hallym University College of Medicine, Seoul 05355, Korea

**Keywords:** lymphatic invasion, prognosis, gastric carcinoma

## Abstract

(Background) The aim of this study was to investigate the prognostic impact of lymphatic invasion in gastric cancer, focusing on survival differences between N stage groups. (Methods) A total of 398 consecutive patients who underwent curative gastrectomy for primary gastric adenocarcinoma from January 2006 to December 2015 were analyzed retrospectively using data from a prospectively collected registry database. We compared various clinicopathological features and survival differences between lymphatic invasion-positive and -negative groups. (Results) Of the 398 patients, 141 (35.4%) showed lymphatic invasion. The lymphatic invasion-positive subgroup had poorer prognosis than the lymphatic invasion-negative subgroup in N0 (five-year survival rate: 87.8% vs. 73.6%, *p* = 0.048) and N1 (87.2% vs. 50%, *p* = 0.007) stage patients. The odds ratio (OR) of lymphatic invasion to five-year survival rate was 2.078 (95% confidence interval (CI), 1.103–3.916; *p* = 0.024). The presence of lymphatic invasion had worse effect on survival than age (OR, 1.807; 95% CI, 1.024–2.242; *p* = 0.029) or tumor depth (OR, 1.286; 95% CI, 1.078–1.897; *p* = 0.013) in N0 and N1 stage patients. The overall survival of patients with lymphatic invasion was not different from that of patients at a one-higher N stage without lymphatic invasion at any N stage. (Conclusions) The presence of lymphatic invasion may be the most important independent prognostic factor in N0 and N1 gastric cancer and might be an upstaging factor of N stage at any N stage. Therefore, in addition to the number of metastasized lymph nodes, the presence of lymphatic invasion should be included in N stage determination.

## 1. Introduction

TNM classification (depth of invasion of tumor (T), lymph node metastasis (N), and distant metastasis (M)) has been known to be an important indicator for prognosis such as survival and recurrence in gastric cancer [1,2]. Since 2009, the Seventh American Joint Committee on Cancer (AJCC) TNM classification has been used for staging, choice of treatment modality, and predicting prognosis in gastric cancer [2]. However, the number of metastatic lymph nodes included in TNM classification has been debated with regard to predicting prognosis [3,4,5]. Therefore, other parameters such as lymph node ratio (ratio of number of metastatic lymph nodes and total harvested nodes) and log odds of metastatic lymph nodes (log of ratio of number of metastatic and negative nodes) were proposed [6,7,8].

Lymphatic channels play a pivotal role in the spread and recurrence of solid organ tumors. Lymphatic invasion (LI) of malignant tumors, acting as a micro-metastatic tumor focus, is one of the useful predictive markers of lymph node metastasis and cancer recurrence in gastric cancer [9,10,11] as well as in various types of cancers such as esophageal cancer [12], colorectal cancer [13], melanoma [14], non-small cell lung cancer [15], and epithelial ovarian cancer [16]. However, the effect of LI on survival in gastric cancer is controversial. Metastatic lymph nodes are found in only about 5–10% of patients with LI in endoscopic resection-treated early gastric cancer (EGC). In fact, we have encountered gastric cancer patients showing discrepancy between LI status and lymph node metastasis [17,18]. Many studies have consistently concluded that LI is a prognostic factor only in a subset of node-negative gastric cancers and may have little prognostic value in patients with lymph node metastasis, even at N1 stage [19,20,21,22]. Therefore, it is necessary to identify additional predictors, other than TNM status, for stratifying patients at risk for poor prognosis. These predictors may have great clinical significance, aiding in patient selection for more extensive surgery and further adjuvant therapies, to improve survival.

Therefore, we assessed the clinicopathological features and prognosis of gastric cancer with lymphatic invasion, with a specific focus on survival differences related to the presence and absence of lymphatic invasion in each N stage group.

## 2. Materials and Methods

### 2.1. Study Population

We included consecutive primary gastric adenocarcinoma patients who underwent subtotal or total gastrectomy from January 2006 to December 2015 using our registry at the Kangdong Sacred Heart Hospital. The inclusion criteria for this study were as follows: (1) primary gastric adenocarcinoma treated with curative gastrectomy and standard lymph node dissection, (2) survival data available. The exclusion criteria were as follows: (1) recurrent gastric cancer, (2) underwent palliative surgery, (3) history of other cancers within five years before or after gastric cancer diagnosis. Of 503 consecutively enrolled patients, 58 underwent palliative surgery and 15 underwent surgery due to recurrent gastric cancer. Survival data was unavailable for 32 patients. Thus, 105 patients were excluded, and a total of 398 patients were included in this study (Figure 1). None of the patients received chemotherapy preoperatively.

### 2.2. Definition

Two well-trained surgeons with extensive experience in gastrectomy performed surgery and perioperative care following the standardized operating procedures and protocols of our hospital, as per the Japanese gastric cancer treatment guidelines [23,24]. There was no difference statistically between the surgical results at each stage in terms of survival rate, as per both the surgeons.

In curative gastrectomy patients, final pathological findings were used for tumor staging based on the Seventh AJCC TNM classification.

Curative resection was defined as gross removal of all cancer masses with demonstration of tumor-free surgical margins and dissection of lymph nodes with optimal extent according to the Japanese gastric cancer treatment guidelines. D1 lymph node dissection plus removal of the lymph nodes along the left gastric artery, the common hepatic artery, and the celiac trunk was performed in EGC patients without enlarged lymph nodes on a preoperative staging abdominopelvic computerized tomography (APCT) scan. All other patients underwent D2 or more lymph node dissection.

In histological classification, the differentiated types included papillary adenocarcinomas and well- and moderately differentiated tubular adenocarcinomas, whereas the undifferentiated types included poorly differentiated tubular adenocarcinomas, mucinous adenocarcinomas, and signet-ring cell carcinomas.

The presence of LI was determined by routine hematoxylin and eosin staining of the resected specimens and immunostaining using an anti-D2-40 antibody as a lymphatic marker.

### 2.3. Study Design

Registry data, including patient demographics, risk factors (e.g., smoking and alcohol consumption), underlying diseases (e.g., hypertension and diabetes mellitus), laboratory findings, preoperative staging work-up results (e.g., computed tomography and endoscopy), histopathological results, cancer stage, and survival, were collected prospectively and analyzed retrospectively.

Patients were divided into LI-positive and -negative groups, and clinicopathological features including age, sex, number of tumors, stomach resection type, location, size of primary tumor, histological classification, and tumor depth and nodal status, were compared. Survival differences between the LI-positive and -negative groups were then assessed, and independent predictors of poor prognosis were identified.

Survival information was obtained from our hospital records and the Korea central cancer registration database at the National Cancer Center. The data of relapse could be analyzed only in the patients who visited our hospital continuously. For patients who failed to undergo continuous observation at our hospital, we obtained survival data from the central cancer registration database.

### 2.4. Ethics Approval

This study protocol was approved by the institutional review boards of the Kangdong Sacred Heart Hospital and meets the guidelines of their responsible governmental agency (IRB file no.: 2017-11-015).

### 2.5. Statistical Analysis

The χ^2^ test for categorical variables and Student’s *t*-test for continuous variables were performed to compare clinicopathological features. Multivariate analyses, using the Cox proportional hazards regression model, were performed to identify independent predictors of prognosis. Overall survival rates were obtained by the Kaplan–Meier method, and differences between the two groups were evaluated by using the log-rank test. All statistical analyses were conducted using SPSS version 24.0 software (IBM Inc., Armonk, NY, USA), and a *p*-value < 0.05 was considered statistically significant.

## 3. Results

Patient baseline characteristics and correlations between LI and clinicopathological features are summarized in Table 1. Of the 398 patients, 141 (35.4%) showed LI. The male-to-female ratio was 2.75:1, and the mean age was 60.4 years (range, 23–87 years). Distal subtotal gastrectomy was performed in 311 patients, and 85 received total gastrectomy. EGC diagnosis was issued to 174 (43.7%) patients and 227 (57%) had no lymph node metastasis (N0) as per the final pathological report. Among mucosa-confined EGCs, there was no LI. In patients with LI, the primary tumor size was significantly larger (5.59 vs. 3.22 cm, *p* < 0.0001). The total gastrectomy group displayed LI more frequently than did the distal subtotal gastrectomy group (*p* = 0.001). The incidence of LI correlated positively with T stage (T1, 13/174, 7.5%; T2, 20/50, 40.0%; T3, 65/102, 65.7%; T4, 41/72, 56.9%; *p* < 0.0001), and N stage (N0, 31/227, 13.7%; N1, 27/57, 47.4%; N2, 28/43, 65.1%; N3, 55/71, 77.5%; *p* < 0.0001). However, other clinicopathological variables such as sex, tumor multiplicity, histological type, and tumor location showed no statistical differences between the two groups.

The median follow-up period was 60.2 months (range, 0.5–119.9 months) and the five-year survival rates (5YSRs) were 90.8% in stage I, 70.9% in stage II, and 51.9% in stage III. As expected, the overall survival rate of the LI-negative group was superior to that of the LI-positive group (5YSRs, 84.0% vs. 55.7%, respectively; *p* < 0.0001) (Figure 2). When comparing the survival rate according to the status of LI for each stage, the stage I LI-positive group tended to have a worse survival rate, but it was not statistically significant (5YSRs, 79.2% vs. 91%, *p* = 0.109). In stage II, the LI-positive group had a significantly poor prognosis (5YSRs, 56.2% vs. 81.9%, *p* = 0.003), and in stage III, there was no difference between the two groups (5YSRs, 48.6% vs. 56.8%, *p* = 0.702). 

When we compared 5YSRs according to LI status in each N stage group, the LI-positive subgroup had a poorer prognosis than did the LI-negative subgroup in N0 (5YSRs, 87.8% vs. 73.6%, *p* = 0.048) and N1 (5YSRs, 87.2% vs. 50%, *p* = 0.007) stages. However, there were no survival differences between subgroups in N2 and N3 stages (Figure 3).

In N0 and N1 stage subgroups, the presence of LI, older age, large tumor size (>5 cm), and T stage were significant factors associated with poor prognosis in univariate analysis (Table 2). Other factors such as sex, tumor multiplicity, tumor location, and histological type were not associated with prognosis. In multivariate analysis, the presence of LI, age, and T stage were identified as independent prognostic factors. The odds ratio (OR) of LI to 5YSR was 2.078 (95% confidence interval (CI), 1.103–3.916; *p* = 0.024), and the presence of LI had more influence on survival than did age (OR, 1.807; 95% CI, 1.024–2.242; *p* = 0.029) or tumor depth (OR, 1.286; 95% CI, 1.078–1.897; *p* = 0.013) in N0 and N1 stage patients (Table 2).

To analyze the effect of LI on N stage, we compared survival rates between LI-positive patients at each N stage and LI-negative patients at a one-higher N stage. Survival curves showed that the 5YSR of N0 patients with LI (N0 LI (+) subgroup) was not different from that of N1 patients without LI (N1 LI (−) subgroup) (73.6% vs. 87.2% of 5YSRs, *p* = 0.366) (Figure 4A). The 5YSR of the N2 LI (−) subgroup tended to be slightly better than that of the N1 LI (+) subgroup, though there was no statistically significant difference between the two groups (66.0% vs. 50.0%, *p* = 0.352, Figure 4B). There was no statistical difference in survival rates between N2 LI (+) and N3 LI (−) subgroups (61.3% vs. 43.8%, *p* = 0.325, Figure 4C).

## 4. Discussion

We demonstrated that LI was an independent prognostic factor for predicting survival, particularly in N0 and N1 stage patients. The 5YSRs were 90.8% in stage I, 70.9% in stage II, and 51.9% in stage III in our data, in agreement with that of other studies [25,26,27], whereas the 5YSRs of the patients with LI were 79.2% in stage I, 56.2% in stage II, and 48.6% in stage III. This result implied that LI negatively affected the survival of patients with gastric cancer [9,11,22,28,29,30]. Because the survival rate differed significantly in N0 and N1 groups according to LI status, we re-examined the factors affecting survival in these patients. Multivariate analysis using the Cox proportional hazard model revealed that the OR of LI was higher than that of T factors (known to be most important prognostic factors in gastric cancer). However, there were no survival differences according to LI status in the more advanced N stage groups such as N2 or N3. It is possibly explained that lymph node metastasis status may wield a stronger influence on prognosis than LI status does. Therefore, LI affects the prognosis in early N stages independently.

Next, we compared survival rates of patients with and without LI in the same N stage, and then compared the survival rates of patients showing LI with those at a higher N stage without LI, to assess the effect of LI on survival. Surprisingly, the 5YSR of patients with LI was similar to that of patients without LI in the one-higher N stage group (N0 with LI vs. N1 without LI (*p* = 0.366), N1 with LI vs. N2 without LI (*p* = 0.352), N2 with LI vs. N3 without LI (*p* = 0.325)). These results suggest that each N stage patient with LI has a poor prognosis similar to one-step higher N stage patients without LI. One previous study reported that the 5YSR was similar between EGC patients without lymph node metastasis (T1N0M0) with LI, and those with lymph node metastasis (T1N1M0) without LI [19]. Thus, the current AJCC staging system may not reflect precise prognosis because this classification does not consider LI status. It might be necessary to account for LI in the current staging system and treat the presence of LI as an upstaging factor of N stage. Furthermore, LI status can be a decision-making factor for extensive dissection of perigastric lymph node or additional chemotherapy after surgery in early N stage patients. Appropriate treatment selection according to tumor status is essential to improve patient survival. Some studies have concluded that node-negative EGCs with LI are associated with poor prognosis and higher risk of recurrence than those without LI and, thus, should receive additional therapy after initial surgery [9,20,22,29].

Although the mechanism of tumor metastasis through lymphatics remains unclear, lymphatic spread of cancer is assumed to occur through cancer cells penetrating into peritumoral lymphatics and reaching the regional lymph nodes. Thus, close contact between cancer cells and lymphatics is thought to be a major step in lymphatic metastasis. Lymphatic vessels are rare at the center of gastric tumors since high interstitial pressure at the tumor center causes collapse and destruction of lymphatic vessels [31]. In contrast, there are abundant lymphatic vessels in the peripheral zone of tumors, especially in the superficial one-third of the submucosal layer (≤500 μm, SM1), and the main source of lymph node metastasis may be the LI in the SM1 layer in gastric cancers extending over the submucosal layer [32]. Actually in this study, there was no LI in 103 mucosal-layer confined EGCs, whereas LI was observed in 18.3% of EGCs involving the submucosal layer.

Two recent advancements in less invasive surgeries involve a reduction of resection size of the stomach or a reduction in the scope of lymphadenectomy. However, LI is closely related to the presence of lymph node metastasis. Absence of lymph node metastasis does not signify absence of lymphatic spread and also cannot negate the value of lymphadenectomy. Therefore, this suggests that LI may be used as an indicator for a more extensive surgical resection. Furthermore, if LI is present or suspected, radical gastrectomy, including removal of proper extent of lymph nodes and complete resection of surrounding tissues may be indicated even in N0 stage patients. In addition, preoperative or postoperative chemotherapy may be beneficial in eliminating micro-metastasis in the lymphatic system.

The strengths of the present study are as follows: First, we investigated the prognostic effect of LI in a large consecutive gastric cancer series at all N stages, with a considerably long follow-up period, using registry data collected prospectively. Second, to minimize bias due to surgical skill and protocol variations, patients who underwent surgery at a single institution were included. Third, accuracy of survival data was ensured since it was obtained from our hospital records and the Korea central cancer registration database at the National Cancer Center.

However, there are some limitations to this study. First, the results require further investigation to reach firm conclusions because of the inherent limitations of a retrospective study. Second, we did not evaluate disease-free survival because disease status of patients lost to follow-up may have been imprecise. However, in Korea, all patients are covered by the national health insurance system, and also every cancer patient is registered in the Korea central cancer registration database. Moreover, since this is the most reliable information, we used it and evaluated overall survival, which is a concrete end point that is clearly observable.

In conclusion, the presence of LI was an important independent prognostic factor in N0 and N1 stage gastric cancers. LI might be a possible upstaging factor of N stage in all stages. We suggest that both the number of metastasized lymph nodes and the presence of LI be included in the TNM gastric cancer staging system.

## Figures and Tables

**Figure 1 jcm-09-01275-f001:**
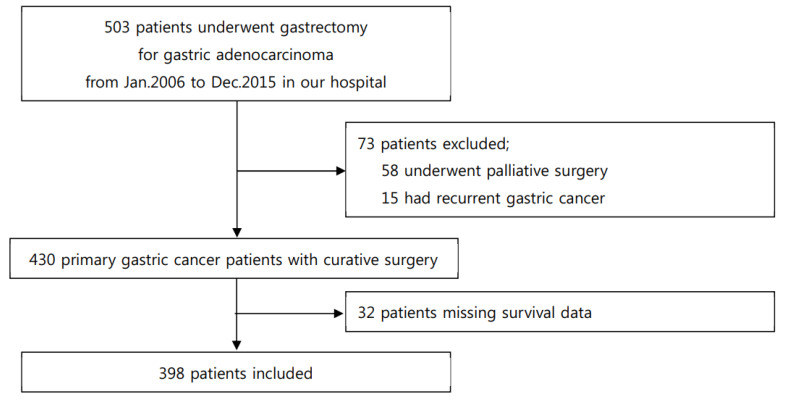
Flowchart of enrolled patients.

**Figure 2 jcm-09-01275-f002:**
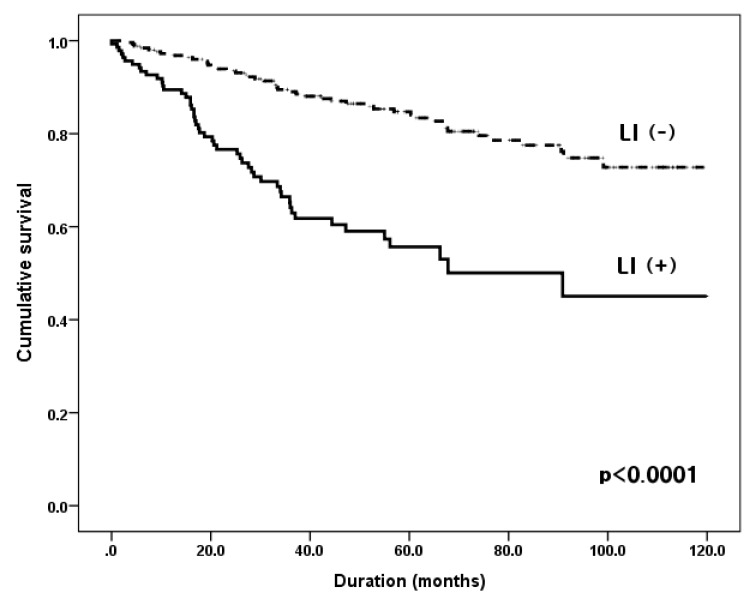
Cumulative overall survival according to the status of lymphatic invasion of 398 patients. Group LI (+), patients with lymphatic invasion; Group LI (−), patients without lymphatic invasion.; five-year survival rates 55.7% vs. 84%, *p* < 0.0001.

**Figure 3 jcm-09-01275-f003:**
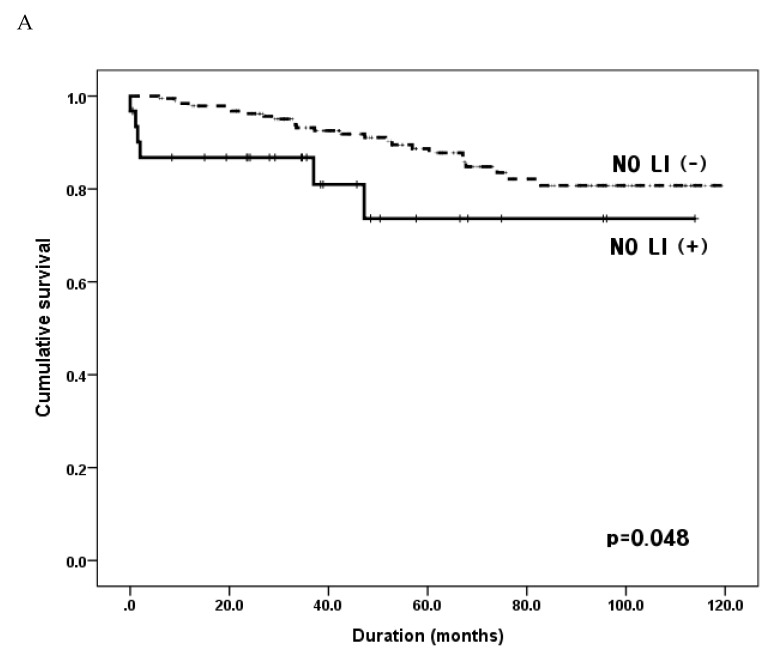
Comparison of overall survival according to the status of lymphatic invasion in each N stage. (**A**) In N0 patients. Group N0 LI (+), N0 patients with lymphatic invasion; Group N0 LI (−), N0 patients without lymphatic invasion; five-year survival rates of 73.6% vs. 88.7%, *p* = 0.048. (**B**) In N1 patients. Group N1 LI (+), N1 patients with lymphatic invasion; Group N1 LI (−), N1 patients without lymphatic invasion; five-year survival rates of 50.0% vs. 87.2%, *p* = 0.007. (**C**) In N2 patients. Group N2 LI (+), N2 patients with lymphatic invasion; Group N2 LI (−), N2 patients without lymphatic invasion; five-year survival rates of 61.3% vs. 66.0%, *p* = 0.564. (**D**) In N3 patients. Group N3 LI (+), N3 patients with lymphatic invasion; Group N3 LI (−), N3 patients without lymphatic invasion; five-year survival rates of 45.1% vs. 43.8%, *p* = 0.669.

**Figure 4 jcm-09-01275-f004:**
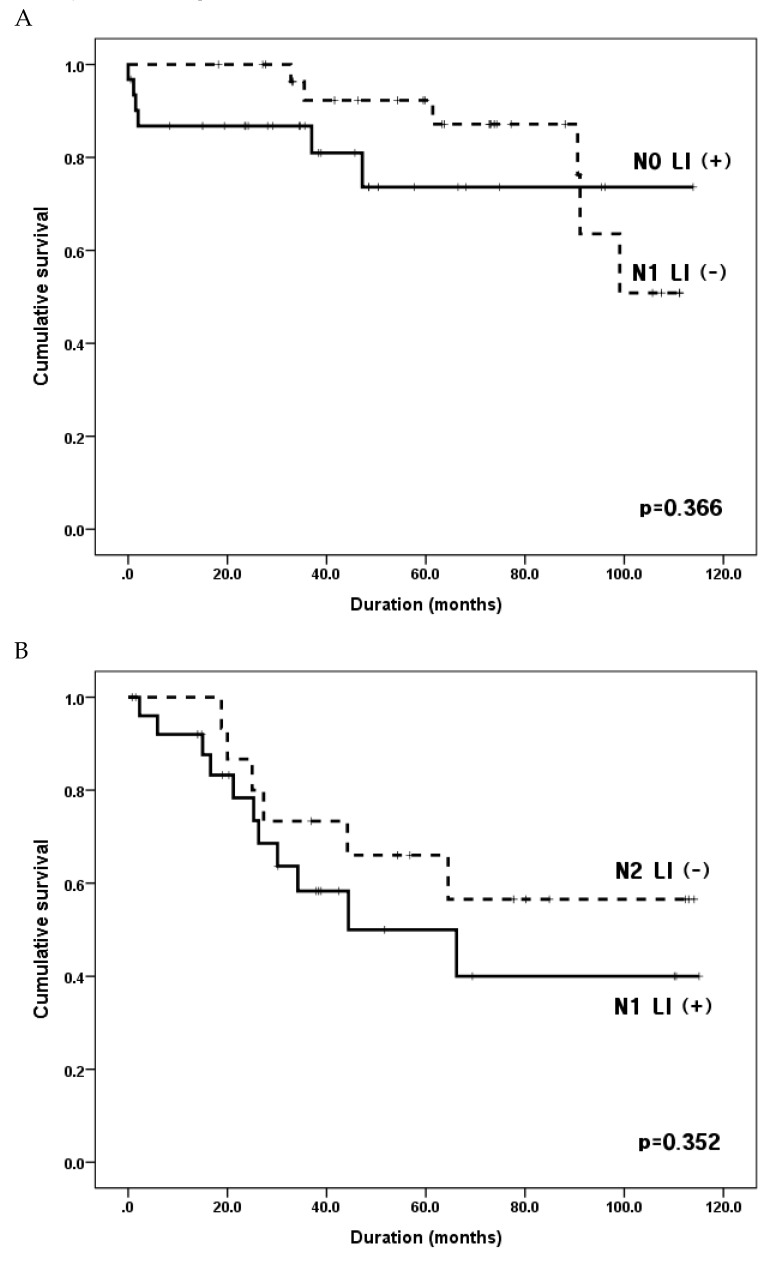
Comparison of overall survival between lymphatic invasion positive patients in each N stage and lymphatic invasion negative patients with one higher N stage. (**A**) Group N0 LI (+), N0 patients with lymphatic invasion; Group N1 LI (−), N1 patients without lymphatic invasion; five-year survival rates of 73.6% vs. 87.2%, *p* = 0.366. (**B**) Group N1 LI (+), N1 patients with lymphatic invasion; Group N2 LI (−), N2 patients without lymphatic invasion; five-year survival rates of 50.0% vs. 66.0%, *p* = 0.352. (**C**) Group N2 LI (+), N2 patients with lymphatic invasion; Group N3 LI (−), N3 patients without lymphatic invasion; five-year survival rates of 61.3% vs. 43.8%, *p* = 0.325.

**Table 1 jcm-09-01275-t001:** Patient characteristics and correlation between lymphatic invasion and clinicopathologic features.

Variables	No. of Patients (%)	Lymphatic Invasion (+) (*n* = 141)	Lymphatic Invasion (−) (*n* = 257)	*p*-Value
Age (years, mean ± SD)		62.49 ± 11.33	59.31 ± 11.55	0.130
Gender				0.073
Male	292 (73.4)	111 (78.7)	181 (70.4)	
Female	106 (26.6)	30 (21.3)	76 (29.6)	
No. of cancer				0.856
Single	371 (93.2)	131 (92.9)	240 (93.4)	
Multiple	27 (6.8)	10 (7.1)	17 (6.6)	
Resection of stomach				0.001
Distal subtotal gastrectomy	311 (78.5)	98 (69.5)	213 (82.9)	
Total gastrectomy	85 (21.5)	43 (30.5)	42 (16.3)	
Other			2 (0.8)	
Tumor size (cm), (mean ± SD)		5.59 ± 3.09	3.22 ± 2.31	<0.0001
Histology				0.697
Differentiated	231 (58.0)	80 (56.7)	151 (58.8)	
Undifferentiated	167 (42.0)	61 (43.3)	106 (41.2)	
Tumor location				0.332
Upper 1/3	68 (17.1)	29 (20.6)	39 (15.2)	
Middle 1/3	77 (19.3)	24 (17.0)	53 (20.6)	
Lower 1/3	253 (63.6)	88 (62.4)	165 (64.2)	
Tumor depth				<0.0001
Mucosa	103 (25.9)	0 (0.0)	103 (40.1)	
Submucosa	71 (17.8)	13 (9.2)	58 (22.5)	
Proper muscle	50 (12.6)	20 (14.2)	30 (11.7)	
Subserosa	102 (25.6)	67 (47.5)	35 (13.6)	
Serosa	72 (18.1)	41 (29.1)	31 (12.1)	
N stage				<0.0001
N0	227 (57.0)	31 (22.0)	196 (76.3)	
N1	57 (14.3)	27 (19.1)	30 (11.7)	
N2	43 (10.8)	28 (19.9)	15 (5.8)	
N3a	47 (11.8)	39 (27.7)	8 (3.1)	
N3b	24 (6.0)	16 (11.3)	8 (3.1)	

Values are presented as number (%). SD, standard deviation.

**Table 2 jcm-09-01275-t002:** Univariate and multivariate analysis of prognostic variables for N0 and N1 stage patients.

		Univariate Analysis			Multivariate Analysis		
Variables		No.	5YSR (%)	*p*-Value	OR	*p*-Value	95% CI
Lymphatic invasion	(−)	226	89.2%	<0.0001	2.078	0.024	1.103–3.916
	(+)	58	63.1%				
Age	≤60	126	92.4%	0.004	1.807	0.029	1.024–2.242
	>60	158	78.2%				
Gender	Male	202	84.9%	0.657			
	Female	82	83.8%				
No. of cancer	Single	264	84.2%	0.389			
	Multiple	20	89.5%				
Tumor size	<5 cm	236	87.4%	0.006	1.286	0.463	0.657–2.517
	>5 cm	48	70.5%				
Histology	Differentiated	171	83.4%	0.580			
	Undifferentiated	113	86.3%				
Tumor location	Upper 1/3	43	84.5%	0.411			
	Middle 1/3	56	86.5%				
	Lower 1/3	185	84.1%				
T stage	T1	172	89.3%	<0.001	1.430	0.013	1.078–1.897
	T2	42	88.7%				
	T3	44	71.2%				
	T4	26	70.7%				

5YSR, five-year survival rate; OR, odds ratio; CI, confidence interval.

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
