# Peer review of "Lymphatic Invasion Might Be Considered as an Upstaging Factor in N0 and N1 Gastric Cancer†"

_jcm, 2020, doi:10.3390/jcm9051275_

Round 1

Reviewer 1 Report

Summary

This retrospective study by Choi et al. investigates the prognostic value of lymphatic vascular invasion (LVI) in low stage (N0 and N1) gastric cancer patients. The concise and robust study demonstrates that LVI is an independent prognostic indicator for the 5 year survival rate and may have a larger impact on patient survival than age, tumor size, or T-stage.

Broad comments

The major weakness of this study is its limited novelty. The prognostic value of LVI in early gastric cancer has been shown previously, both in patients with (e.g. Li et al, BMC Cancer 2015) and without lymph node metastasis (e.g. Lee et al, World J Surg 2015). Also, the prognostic value of LVI specifically in N0 gastric cancer patients has recently been addressed (Lu et al, BMC Cancer 2019). Although I do recognize the value of complementary clinical studies, the impact of this study is limited.

Specific comments

- Does LVI correlate with lymphatic vessel density in this patient cohort?

- Were there any correlations between LVI and the other risk factors (smoking, alcohol consumption etc.) included in the dataset?

- For comparability with other studies, could you report the disease-free survival at least for those patients for whom follow-up data was available?

- The authors should consider changing the title to an affirmative statement.

Reviewer 2 Report

The submitted manuscript reports data about the fundamental importance of lymphatic invasion in gastric cancer patients, comparing such clinical evidence with the N stage. As the author stated, there is the urgency to improve the TNM system which not always reflects the real status of gastric cancer patients. I do believe that the manuscript fully accomplish the task that authors posed and being well written and reporting many convincing clinical evidences.

Reviewer 3 Report

The authors have attempted to elucidate the role of LI as a prognostic marker in gastric cancer patients. 

Mortality data should be reported with much better granularity among each groups and with respect to TNM staging. 

In the demographics data table, the total number of patients included were 398 but in the multivariate table the number of patients included has decreased significantly to almost 2/3 rd of the original population. What is the reason?

Further in the multivariate table each variable has different total number of patients for analysis. Kindly address this. 

The original number of LI + patients were 141 but in the univariate and multivariate analysis, the total number of patients with LI + were 58?? I am afraid dropping more than 50 to 60% of population will skew the results. If they have missing data, kindly omit the patients from entire analysis. Recommend redoing the statistical analysis with appropriate patient exclusion. 

Tables: Please provide the relative percentages within the LI negative and positive groups and not across the groups which makes it difficult to compare. Highly recommend to change this for ease of understanding by the readers. 

Median months of FU is 60.2 (so roughly half of the population had fu less than 5 years) and thus any information on risk of recurrence is lagging which might play a crucial role on survival, especially a significant number of patients with LI underwent total gastrectomy as opposed to LI negative patients. This variable should have been included in the regression model as this affects survival and statistically different between the groups. 

Further why lymph node staging (N) was not included in the univariate/or multivariate model? This is crucial as they are independent predictors of survival.

Round 2

Reviewer 1 Report

I don't have further comments

Reviewer 3 Report

Authors have successfully answered the queries and shall be accepted.